# Choice of vector and surgical approach enables efficient cochlear gene transfer in nonhuman primate

Eva Andres-Mateos[1,2,12,14], Lukas D. Landegger [3,4,13,14], Carmen Unzu[1,2,14], Jean Phillips[4], Brian M. Lin[4], Nicholas A. Dewyer [4], Julio Sanmiguel[1,2], Fotini Nicolaou[1,2], Michelle D. Valero[3,4], Kathrin I. Bourdeu[5], William F. Sewell[3,4], Rudolph J. Beiler[6], Michael J. McKenna[3,4,7,8,12,15✉], Konstantina M. Stankovic [3,4,8,9,10,15✉] & Luk H. Vandenberghe [1,2,9,11,15✉]

Inner ear gene therapy using adeno-associated viral vectors (AAV) promises to alleviate hearing and balance disorders. We previously established the benefits of Anc80L65 in targeting inner and outer hair cells in newborn mice. To accelerate translation to humans, we now report the feasibility and efficiency of the surgical approach and vector delivery in a nonhuman primate model. Five rhesus macaques were injected with AAV1 or Anc80L65 expressing eGFP using a transmastoid posterior tympanotomy approach to access the round window membrane after making a small fenestra in the oval window. The procedure was well tolerated. All but one animal showed cochlear eGFP expression 7–14 days following injection. Anc80L65 in 2 animals transduced up to 90% of apical inner hair cells; AAV1 was markedly less efficient at equal dose. Transduction for both vectors declined from apex to base. These data motivate future translational studies to evaluate gene therapy for human hearing disorders.

[1] Grousbeck Gene Therapy Center, Schepens Eye Research Institute and Massachusetts Eye and Ear, Boston, MA 02114, USA. [2] Ocular Genomics Institute, Department of Ophthalmology, Harvard Medical School, Boston, MA 02114, USA. [3] Eaton Peabody Laboratories, Massachusetts Eye and Ear, Boston, MA 02114, USA. [4] Department of Otolaryngology – Head and Neck Surgery, Massachusetts Eye and Ear and Harvard Medical School, Boston, MA 02114, USA. [5] Department of Anesthesiology, Massachusetts Eye and Ear, Harvard Medical School, Boston, MA 02114, USA. [6] Animal Science Center, Boston University, Boston, MA 02118, USA. [7] Otopathology Laboratory, Massachusetts Eye and Ear, Boston, MA 02114, USA. [8] Speech and Hearing Bioscience and Technology Program, Harvard University, Cambridge, MA 02138, USA. [9] Harvard Stem Cell Institute, Harvard University, Cambridge 02138 MA, USA. [10] Department of Otolaryngology – Head and Neck Surgery, Stanford University School of Medicine, Stanford, CA 94305, USA. [11] The Broad Institute of Harvard and MIT, Cambridge, MA 02142, USA. [12] Present address: Akouos Inc., 645 Summer St, Suite 200, Boston, MA 02210, USA. [13] Present address: Department of Otolaryngology, Vienna General Hospital and Medical University of Vienna, Vienna 1090, Austria. [14] These authors contributed equally: Eva Andres-Mateos, Lukas D. Landegger, Carmen Unzu. [15] These authors jointly supervised the work: Michael J. McKenna, Konstantina M. Stankovic, Luk H. Vandenberghe. ✉email: mmckenna@akouos.com; kstankovic@stanford.edu; luk_vandenberghe@meei.harvard.edu

Hearing loss is the most common sensory deficit, currently disabling nearly 500 million people[1]. In the United States, around 3 in 1000 are diagnosed with hearing loss at birth[2] and, between the ages of 65 and 74, approximately one in three people suffers from it[3]. The most common type of hearing loss is called sensorineural hearing loss (SNHL) and it typically originates from damage to the delicate mechanosensory structures within the inner ear. The inner ear consists of the cochlea and vestibular end organs. Within the cochlea, there are two types of hair cells, inner hair cells (IHCs), which transform mechanical vibration into electrical signals transmitted to the brain via spiral ganglion neurons (SGNs), and outer hair cells (OHCs), which amplify and tune the cochlear response to sound.

Currently, treatment options for SNHL are limited to hearing aids, cochlear, auditory brainstem, and middle ear implants. While these treatments are often effective, they are unable to restore natural hearing perception. Depending on the etiology, pharmacological and biological therapies would have the potential to halt, reverse, or limit forms of hearing loss. However, small molecular and protein drugs often have poor access and biodistribution to the cochlea, limiting their efficacy. A local route of delivery is thus preferred.

Conceptually, a local injection of a therapeutic gene agent to the inner ear is attractive, as it allows for a single administration of a gene construct that has a durable therapeutic effect. From a safety perspective, it overcomes the need for multiple injections in the hard to access and sensitive cochlear environment. In addition, the local route of administration limits the dose and biodistribution of the agent and a partial immune-privileged status due to the blood-labyrinth barrier of the inner ear may limit noxious anti-drug inflammatory responses. Moreover, 80% of prelingual deafness is genetic and more than 120 causative genes have been identified[4,5]. For recessive disorders, gene augmentation approaches have previously been shown to work in the inner ear in animals. By using different mouse models, functional rescue and thus therapeutic correction of hearing loss and vestibular disorders could be demonstrated[6–15].

Different types of viral vectors have been considered for gene delivery in animal models[6,16–18]. For many different therapeutic target cells and tissues, replication-defective vectors based on the adeno-associated virus (AAV) combine strong transduction efficiency and favorable safety in vivo. Naturally occurring AAVs are not associated with any disease in humans, and they are able to transduce post-mitotic cells, including neurons. The recombinant AAV vectors used for gene therapy maintain their genome in episomes in non-dividing cells and allow for long-term expression[19]. Based on data from a panoply of available AAV capsids, the proteinaceous shell that protects a single-stranded DNA genome is a major determinant of its targeting, immunological, and other properties. These attributes continue to justify the increasing utility of AAV in clinical gene therapy applications across various fields and the approval of two AAV-based drug products[20].

Several AAV capsid variants have been effectively used to perform gene delivery to the murine inner ear at various ages[21–27]. One serotype that showed high efficiency targeting inner and outer hair cells and other cochlear and vestibular cells in neonatal and adult mice was Anc80L65[22,24]. Anc80L65 was derived *in silico* through ancestral sequence reconstruction and is a putative common ancestor of AAV1, 2, 8, and 9[28]. Anc80L65 has been extensively characterized as a potent gene transfer vector in murine liver, muscle, inner ear, retina and kidney[22,24,28–30], and nonhuman primate (NHP) liver and muscle[28].

Intracochlear delivery of AAV can be achieved through a variety of direct surgical approaches[31]. Many of these are used for routine otologic indications and include surgical drilling of a cochleostomy adjacent to the round window, application through or over the round window membrane itself, the creation of a fenestra in the stapes footplate or a semicircular canal.

Unlike rodent models, the NHP cochlea has similar anatomy and proportions to humans. While the surgical approach between NHP and humans is expected to be distinct, the fluid dynamics within the cochlea can be modeled more accurately as it relates to the injection procedure and the distribution of vector within the organ. In addition, given their immunological and metabolic proximity to humans, NHP models are important to assess toxicity and pharmacology of the vector systems as a key translational milestone prior to consideration for human studies.

The objective of this study was (1) to optimize the cochlear injection procedure via the round window membrane (RWM) in the NHP, (2) to gauge the feasibility and tolerability of the surgical route and administration of a fixed, moderate dose of AAV locally and systemically, and (3) to assess the cell tropism and transduction efficiency of Anc80L65 and AAV1 serotype following a RWM injection.

In this work we demonstrate that Anc80L65 allows efficient cochlear gene transfer in nonhuman primates, and encourages future translational studies to evaluate the prospect of gene therapy for hearing disorders.

## Results

**Vector design and validation in mouse model**. AAV constructs were designed to include an optimized Chicken β-actin promoter with early CMV-enhancer (CB7) and chimeric intron (CI) given its prior use in NHP and human studies[32–34]. Due to the novelty of the construct and the promoter combination in the cochlea, prior to injection in NHPs, AAV2/1.CB7.CI.eGFP.RBG and AAV2/Anc80L65.CB7.CI.eGFP.RBG were tested in wild-type mice by systemic and cochlear injection.

In line with expectations for AAV1 and Anc80L65, differential levels of enhanced green fluorescent protein (eGFP) expression were detected in the murine livers with both vectors after intravenous delivery (Supplementary Fig. 1a)[28]. To determine whether the CB7 promoter contributed to a different tropism or cellular expression compared to the previously used CMV promoter[22,24], neonatal mouse cochleae were injected with CMV-driven eGFP and CB7-driven eGFP vectors. No differences were found in the expression or tissue distribution (Fig. 1).

**Development of surgical injection method**. The surgical procedure was developed on cadaveric macaque heads. These dissections revealed that it was not feasible to utilize a transcanal approach as the external auditory canal is too small to allow introduction of surgical instruments, including the smallest available endoscope. We evaluated a transmastoid posterior tympanotomy approach, which achieved excellent exposure of the round window, and exposure of the stapes.

Based on the optimizations performed, the eventual procedure can be described as a transmastoid facial recess exposure of the RWM with round window injection following oval window venting. Specifically, a left postauricular incision was made and dissection of the soft tissue was performed down to the level of the periosteum. The periosteum was incised and elevated to expose the mastoid bone. A cortical mastoidectomy was performed with a combination of high-speed cutting and diamond drill burs (Fig. 2a, b and Supplementary Movie 1). The facial recess was then opened, allowing for adequate visualization of the round window and oval window. Fenestration of the oval window was performed using a Rosen needle (Fig. 2c and Supplementary Movie 1), because we hypothesized that this would allow potential fluid displacement and enable an improved

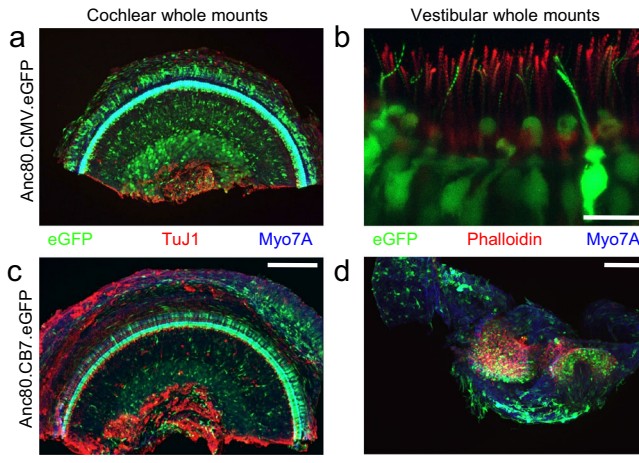

**Fig. 1 eGFP cochlear and vestibular expression after transduction with vectors with the same transgene driven by CMV or CB7 promoter.**
Representative in vivo transduction in cochlear and vestibular whole mounts of CD1 mice 14 days after P4 ipsilateral round window membrane delivery of Anc80L65 using the same eGFP (green) transgene driven by CMV (**a**, **b**) or CB7 (**c**, **d**). Neonatal mouse ($n = 3$ mice per condition) cochleae were injected with CMV- or CB7-driven eGFP vectors at a dose of $2 \times 10^9$ GC per cochlea. Similar eGFP expression and tropism were observed between the two viral vectors. (**a**, **c**) Whole mounts of cochlear middle turns were co-stained with Myo7A (blue, for hair cells) and TuJ1 (red, for neuronal structures). Scale bar, 200 μm for a and c. (**b**, **d**) Whole mounts of vestibular tissue were co-stained with Myo7A (blue, for hair cells) and Phalloidin (red, actin). Two images of cristae ampullares of different animals obtained for CMV with 63x (**b**) and for CB7 with 20x (**d**) objectives. Scale bar, 50 μm for b and 100 μm for d.

vector flow toward the apex of the cochlea after the injection of a larger volume without damage of the internal cochlear structure. Vector injection through the RWM was then performed (Fig. 2d and Supplementary Movie 1). The soft tissue was closed in layers with buried, interrupted, 3-0 monocryl sutures (Ethicon). The skin was re-approximated with running subcuticular 4-0 monocryl suture (Ethicon).

**Procedure and in-life phase.** To evaluate the transduction efficiency in the cochlear sensory epithelium and auditory neurons, we sought to inject one ear of two animals with AAV1 and one ear of three additional animals with Anc80L65 (Table 1). Rhesus macaque was selected as a species given their established use in gene therapy, their genetic, metabolic, and immunological proximity and relevance to human applications, and the size of the cochlea, which is approximately 30% of the human inner ear[35,36].

Three- to four-year-old, female animals were enrolled in the study based on a clean health record, no experimental treatment history, and serum neutralizing antibody titers against AAV1 or Anc80L65 below 1:10.

The study was designed to navigate several unknowns, given the limited precedent of NHP cochlear injections, particularly of an AAV, including dose, volume, potential for inflammatory responses to vector and/or transgene that may harm the animal and the interpretation of the histology, in vivo surgical feasibility, and kinetics of expression. The study design staggered the injections over multiple days to assess surgical feasibility and short-term tolerability. The endpoint of the study was 1 week upon injection to mitigate noxious effects of any potential immunological adaptive responses. One animal dosed with Anc80L65 was monitored for 14 days following injection to

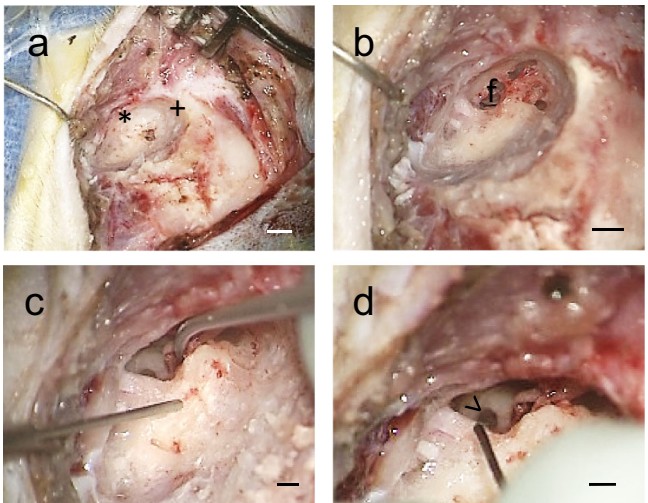

**Fig. 2 Representative images of the surgical procedure in a rhesus macaque's left ear. a** Mastoidectomy was performed to identify the posterior wall of the external auditory canal anteriorly (asterisk) and the tegmen mastoideum superiorly (plus sign). **b** Extended facial recess (f) approach was used to expose the round window membrane after skeletonizing the facial nerve and cutting the chorda tympani nerve for access. **c** A 2-mm fenestration in the oval window was performed. **d** AAV vector was microinjected through the round window membrane (greater-than sign) using a pump. Scale bars, 5 mm in a/b and 1 mm in c/d. See also Supplementary Movie 1.

preliminarily assess the kinetics of expression and whether longer-term expression was tolerated. All animals received an absolute dose of $2.5 \times 10^{11}$ vector genome copies (GC) in 30 μl buffered saline (at a concentration of $8.33 \times 10^{12}$ GC/ml).

All the animals successfully recovered from the surgical procedure with minor physical impairments or discomfort expected after anesthesia and/or ear surgery (imbalanced, scratching of incision site, or decreased activity) that resolved within the first 24 to 48 h. No changes in food and water intake were observed during the in-life phase of the study. Animal RA3131 however developed unilateral facial paresis ipsilateral to the injection with incomplete left eye closure (no lower facial paralysis was observed) a day following surgery. The animal was preventively treated with a lateral tarsorrhaphy 24 h after the procedure. No signs of corneal ulcer, inflammation, or irritation were detected in this animal during the in-life phase of the study. Animals were closely monitored after the procedure for surgical complications; no obvious adverse effects or signs of acute toxicity related to the injection of AAV1 or Anc80L65 viral vectors were observed (Supplementary Table 1).

**Cochlear tropism and transduction efficiency.** Cochleae were harvested seven days after the surgery except for animal RA3128, for which the endpoint was 14 days after the procedure. AAV transduction was evaluated by immunohistochemistry using an anti-GFP antibody. eGFP expression in the IHCs was detected in four monkeys; no expression in any cell type was detected in RA3120, which is why it is not depicted in the Figures. While it was not quantified, sporadic outer hair cell damage was observed in injected and contralateral uninjected ears for all the animals. Inner hair cells were quantified at each frequency (a total of 20 cells per frequency region along the length of the cochlea) and no cell damage was observed with either serotype (data non included). Minimal tissue damage at the most basal portion of the cochlea was detected in a few specimens, most likely related with

**Table 1 Experimental design and neutralizing antibody titers detected in serum and cerebrospinal fluid.**

| Animal ID | Age (yrs) | Weight (kg) | AAV serotype injected | Duration of study (days) | Serum pre-surgery | Serum (day 0) | Serum (end-point) | CSF (end-point) |
|---|---|---|---|---|---|---|---|---|
| | | | | | Neutralizing antibodies against AAV1 | | | |
| RA3009 | 3.7 | 4.70 | AAV1 | 7 | ND | ND | 1/64 | ND |
| RA3109 | 3.7 | 4.45 | AAV1 | 7 | ND | ------- | 1/512 | ND |
| | | | | | Neutralizing antibodies against Anc80L65 | | | |
| RA3120 | 3.6 | 4.20 | Anc80L65 | 7 | ND | ND | 1/64 | ND |
| RA3131 | 3.6 | 4.60 | Anc80L65 | 7 | 1/4 | 1/4 | 1/256 | ND |
| RA3128 | 3.6 | 4.65 | Anc80L65 | 14 | 1/4 | 1/4 | ≥1/4096 | ND |

Luminescence was normalized against control cells transduced with AAV incubated without serum. A neutralizing titer was determined at the dilution at which luminescence was <50% compared to control wells. *ID* animal identification number, *CSF* cerebrospinal fluid.

the specimen post-mortem dissection or the injection procedure performed. No eGFP signal was observed in the histological samples of the contralateral cochlear tissue in any of the treated animals (Supplementary Fig. 1B).

Frequency mapping of the injected cochleae was carried out using images acquired with an epifluorescence microscope equipped with a 10x objective. To evaluate the transduction efficiency in hair cells and supporting cells (SCs), 63x images of each mapped frequency region were taken with a confocal microscope and quantified (Fig. 3a, b). eGFP-positive hair cells were counted, and values were plotted as a percentage of the total number of hair cells. The highest number of cells expressing eGFP was found at the apex, while the lowest number of eGFP-positive cells was observed at the base of the cochlea (Fig. 3a). Transduction efficiency of IHCs by Anc80L65 and AAV1 was different; high transduction of the IHCs was detected for Anc80L65 in all cochlear regions (Fig. 3b). However, only minimal outer hair cell (OHC) transduction was observed for both serotypes when evaluating cochlear whole mounts using confocal microscopy. Specifically, one positive outer hair cell was detected in the 32.0 kHz region of ear RA3009 and three appeared in the 22.6 kHz region of ear RA3128. Overall, no transduction of the SGNs and auditory nerve sections from different regions of the cochlea was detected (Fig. 4). eGFP signal in the modiolus was only observed in one small area of an apical section from animal RA3109 that was injected with AAV1.

In addition to the IHCs, both serotypes were able to transduce SCs (Supplementary Fig. 2). While the GFP fluorescence intensity was lower in the SCs than the IHCs, a large number of SCs were transduced in the animals included in the study. The SCs contribute to the development, structural maintenance, and function of the hair cells; some mutations in genes expressed in these cells result in deafness[37]. A qualitative analysis of the number of eGFP-positive SCs was performed to assess the transduction efficiency (Supplementary Fig. 2a). AAV1 and Anc80L65 transduced SCs at similar efficiency from apex to base. More transduction was detected in the apical region encoding lower frequencies, with a decrease of signal towards the base of the cochlea with both AAV1 and Anc80L65 (Supplementary Fig. 2b).

**Transduction of the vestibular organ and cochlear lateral wall**. The peripheral vestibular system and cochlear lateral wall are relevant targets for gene transfer applications. For example, loss of type II fibrocytes in the cochlear lateral wall as well as degeneration of stria vascularis and spiral ligament contribute to noise-induced hearing loss[38]. A qualitative analysis of the eGFP expression in whole mounts of the vestibular organ is shown in Fig. 5a, b. Only animal RA3131, which received Anc80L65 showed substantial eGFP expression throughout the

peripheral vestibular system; the fluorescent signal was detected in utricle, saccule, ampulla, and membranous semicircular ducts (Fig. 5a, b). Weak fluorescent signal was detected in small areas of the ampullae, vestibule, and saccule of the animals injected with AAV1, RA3009 and RA3109. No expression was detected in animal RA3128, despite the fact that this animal had been injected with Anc80L65 and showed high eGFP signal in the cochlea. Therefore, no direct correlation was found between serotype and transgene expression in the vestibular system or other areas of the inner ear.

The lateral wall was dissected in 7 sections, and whole mounts were examined. A qualitative evaluation of the eGFP expression in the 7 sections of the lateral wall was performed in all the specimens obtained from the four animals where we found any eGFP-positive cells in the inner ear. eGFP signal was detected in at least 3 to 4 sections of the lateral wall of all NHPs examined. Signal distribution and intensity were similar between all animals, and no differences were noticed between serotypes (Fig. 5c). Microglia was stained using an anti-Iba1 antibody, transduction of the microglia (colocalization between eGFP and Iba1 signal) was not observed with either Anc80L65 or AAV1.

**Systemic neutralizing response to AAV**. Neutralizing antibody (NAB) titers against AAV capsid can be developed in humans and NHP as a result of exposure to the wild-type virus or after administration of AAV vectors. The presence of pre-existing NAB can preclude AAV transduction when the vector is delivered by systemic or a local route of administration. Hence, antibody titers against AAV1 and Anc80L65 in serum were determined by neutralization assays at multiple time points - prior to animal purchase, following arrival to the Animal Facility, the day of surgery (Day 0) and at the endpoint (Day 7 and Day 14, respectively). In addition, antibody titers in cerebrospinal fluid (CSF) were determined at the endpoint only. Animals were negative for NAB or showed low antibody titers prior to the surgery, but all the NHP seroconverted against the serotype injected after the surgery (Table 1). Titers were higher when animals were injected with Anc80L65, but there was no direct correlation between the titers and the levels of eGFP expression. No neutralizing antibodies were detected in CSF for either serotype (Table 1).

**Acute vector tolerability and biodistribution after cochlear injection**. In addition to efficient gene transfer, the use of viral vectors for clinical applications requires evaluation of the tolerability profile for future therapeutic use. To evaluate systemic toxicity or inflammatory processes, hematoxylin and eosin staining was performed in different organs and several regions of the central nervous system and a postmortem histopathology analysis was completed for both serotypes (Supplementary Fig. 3). No morphological or pathological changes were detected, and no

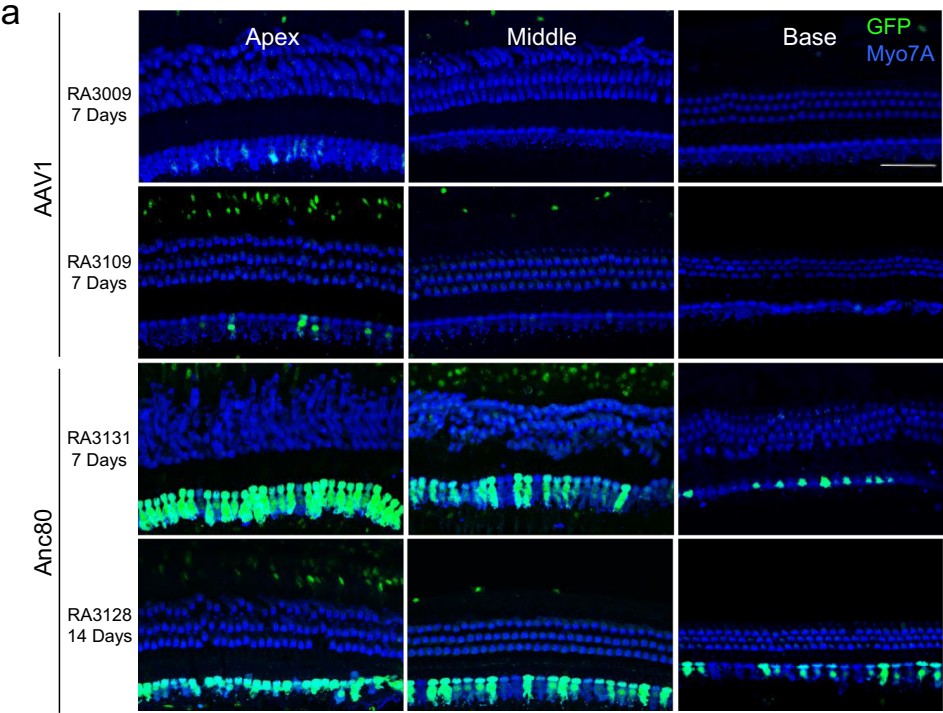

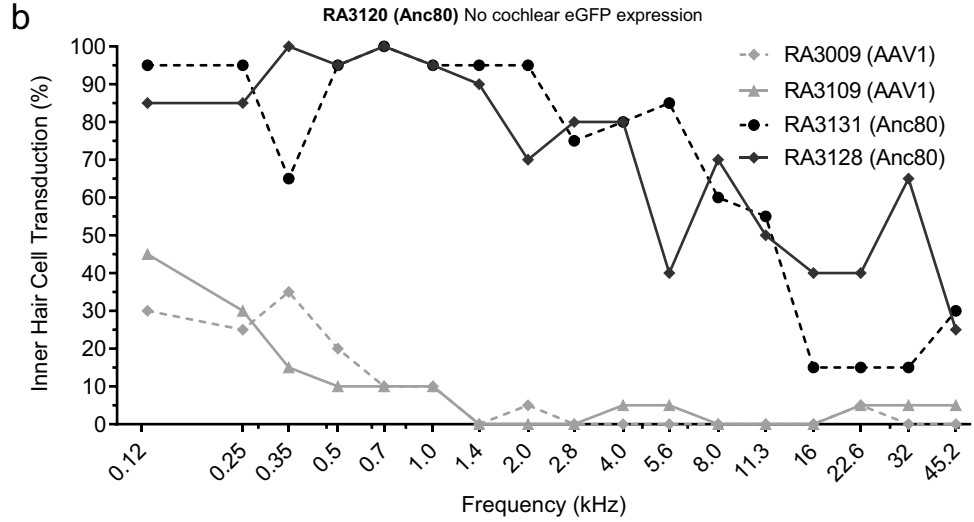

**Fig. 3 In vivo transduction of cochlear cells with AAV1 and Anc80L65 in rhesus macaques. a** Confocal representative images ($n = 2$ for AAV1 and $n = 2$ for Anc80L65) of whole mounts of the organ of Corti areas along the length of the cochlea for all injected inner ears that expressed eGFP (green). Animals were injected with 30 μl of viral vector, total dose of $2.55 \times 10^{11}$ GC. Blue, Myo7A-stained hair cells. Scale bar, 50 μm. No expression in any cell type was detected in RA3120 (Anc80). **b** Quantification of inner hair cell transduction following round window membrane (RWM) injection. Percentage of eGFP-positive inner hair cells (IHCs) per frequency region along the length of the cochlea. Identical confocal microscope settings were used to obtain all images. Source data are provided as a Source Data file.

signs of inflammation or fibrosis were detected in animals injected with AAV1 or Anc80L65.

In addition, animals were phlebotomized pre-surgery (Day 0) and at the endpoint (Day 7 and Day 14) and blood was analyzed for Serum Chemistry (Supplementary Table 2). No evidence of toxicity was observed in the blood chemistry profiles of NHPs treated with AAV1 or Anc80L65. Values were mostly within the normal species-specific reference range, and although a few differences were detected in some parameters, these alterations had no clinical significance.

Finally, biodistribution and shedding of vector genomes in CSF, cerebellum, several cerebral lobes, spinal cord, and liver were determined by droplet-digital PCR (Fig. 5d and Supplementary Table 3). Genome copies detected in cerebellum, cerebral lobes, and spinal cord for both serotypes were considered negative since levels were around background level. Higher vector DNA copies were detected in CSF and liver in animals injected with AAV1 compared with Anc80L65 monkeys, with the exception of animal RA3120, which was injected with Anc80L65 but did not express eGFP in the cochlea.

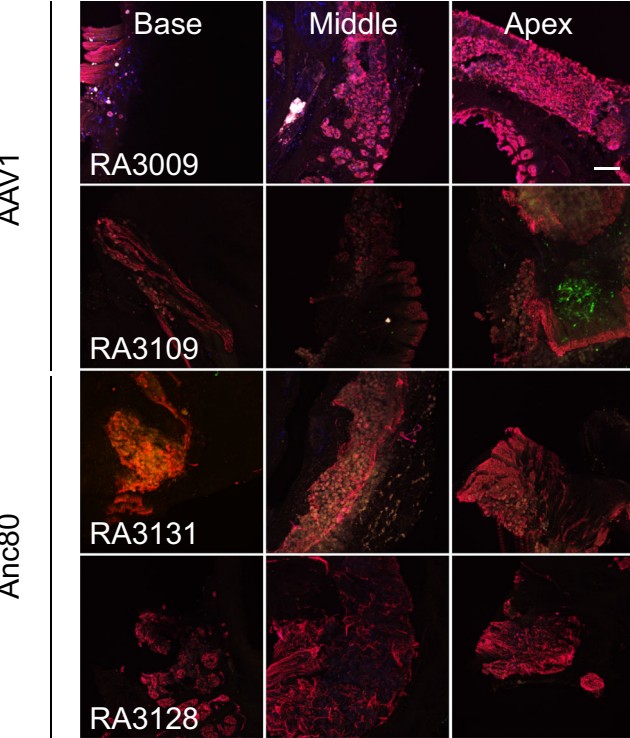

**Fig. 4 In vivo transduction of the spiral ganglion with AAV1 and Anc80L65 in rhesus macaques.** Representative images (from 4 of 5 animals) from the basal, middle, and apical region of the spiral ganglion from each animal reveal hardly any transduction. eGFP (green) signal in the modiolus was only observed in one small area of an apical section from animal RA3109 that was injected with AAV1. Same confocal settings were used to obtain all images. Red, neurofilament-labeled neuronal structures. Scale bar, 50 μm.

Overall, we did not identify any signs of systemic acute toxicity or inflammation related to viral vector injections, and both serotypes showed favorable profiles.

## Discussion

AAV viral vectors are emerging as attractive vehicles for gene targeting in the inner ear due to their satisfactory safety profile and the ability to transduce various cell types in the cochlea. However, the characterization of cell tropism and transduction efficiency for different AAV serotypes and variants in large animal models is essential for the clinical development of successful therapies. Despite the extensive information using various AAV serotypes in the rodent cochlea[21–24,39], there are limited and variable pieces of information available about the efficiency of AAV delivery in large preclinical models[11,40]. Namely, Gyorgy et al.[11] included data of one male cynomolgus monkey with 92% of IHCs and OHCs eGFP transduction seven weeks after delivery of $3 \times 10^{11}$ vector genomes (VG) of AAV9-PHP.B.CBA.eGFP in 10 μL. In contrast, a female macaque injected bilaterally with the same vector and same volume at dose $1 \times 10^{11}$ VG, showed limited cochlear transduction eight weeks after dosing. In both animals, the vector was delivered by injection through the round window membrane. An additional three animals were injected with the same vector mentioned above by bilateral intracochlear injection with three different doses[40]. According to the authors, nearly all of IHCs and OHCs were transduced from base to apex at the two higher doses (3.5 and $7 \times 10^{11}$ VG), while there was a steep reduction in vector transduction at the lower dose ($2 \times 10^{11}$ VG.) Nevertheless, only cochlear cross-sections were examined

and quantified in these studies, which do not allow a direct and equal comparison with our cochlear whole mount data quantification. Expanding on these studies, the same group recently injected three cynomolgus monkeys with a new capsid variant AAV-S[41]. Although the two highest doses (4.7 and $5.8 \times 10^{11}$ VG) showed extensive levels of eGFP expression in a variety of cell types in the NHP cochlea and vestibular system, unfortunately, the analysis in NHP was limited to the AAV-S variant. Hence, comparisons cannot be made with other AAV capsids to more extensively characterize them in the cochlea.

Our results reveal that AAV1 and Anc80L65 can be delivered into the inner ear, and consistently transduce cochlear and vestibular cells of young rhesus macaques using an optimized surgical approach that allows visualization of the round window prior to injection. We demonstrate transduction of IHCs, SCs, cochlear cells within the lateral wall and cells within the peripheral vestibular organs. While most researchers focus on transduction of IHCs and OHCs, our comprehensive examination of additional transduced cell types in the inner ear is relevant for future translation of gene therapies as different genetic diseases affect different auditory and vestibular cell types.

Although both vectors used in this study showed similar levels of eGFP expression in SCs, Anc80L65 was more efficient in transducing IHCs. Unexpectedly, given that the injection was performed in the base of the cochlea, the apical IHCs showed higher eGFP expression levels than the base with a gradient of expression from apex to base ranging from 90–30% for Anc80L65 and 30–5% with AAV1 across the cochlea. This gradient may be due to morphological and metabolic differences in SCs[42], tectorial membrane[43,44] and IHCs[45,46]; future studies beyond the scope of the current paper are required to understand the observed gradients in transduction efficiency. Moreover, the current work studies the cochlear transduction capabilities of AAV1 and Anc80L65 based on the characteristics of the capsid, but transduction efficiency and tropism may be impacted by cell-specific transgene expression or length of coding sequences packaged in AAV[12,14,15].

While the available literature has examined the effects of vector injection at the cochlear level, we also quantified vector central nervous system (CNS) and systemic biodistribution, which is particularly relevant prior to entering human clinical trials. In previous murine studies, a patent cochlear aqueduct was observed and local AAV injection resulted in transduction of cerebellar Purkinje cells[22]. While the patency of the cochlear aqueduct in NHPs and humans has been debated for a long time[47], the majority of adult human temporal bones appear to be occluded with loose connective tissue[48]. However, this does not preclude the possibility of the viral vectors reaching the CNS after cochlear delivery, because some studies concluded that the cochlear aqueducts are patent in humans[49,50]. Our study did not detect any genome copies in the CNS tissue for both serotypes, and no eGFP expression was detected in the contralateral uninjected ear. Evidence of variable genome copies per cell for AAV1 and Anc80L65 in liver and spleen is expected given that these two organs are involved in immune clearance of viruses, and previous studies with Anc80L65[28,30,51] and AAV1[52–54] showed tropism for liver and spleen. The variability in the level of copy numbers in these two organs and CSF may be related to serotype-specific tropism and biodistribution, or variability during the delivery.

Another important aspect of the current study was the optimization of surgical access to the inner ear in preparation for a future human clinical trial. Complementary to existing publications[35], the small fenestration we introduced in the oval window may prevent any cellular mechanical damage due the increase of volume in the scala tympani after the injection. Although no eGFP expression was detected in monkey RA3120,

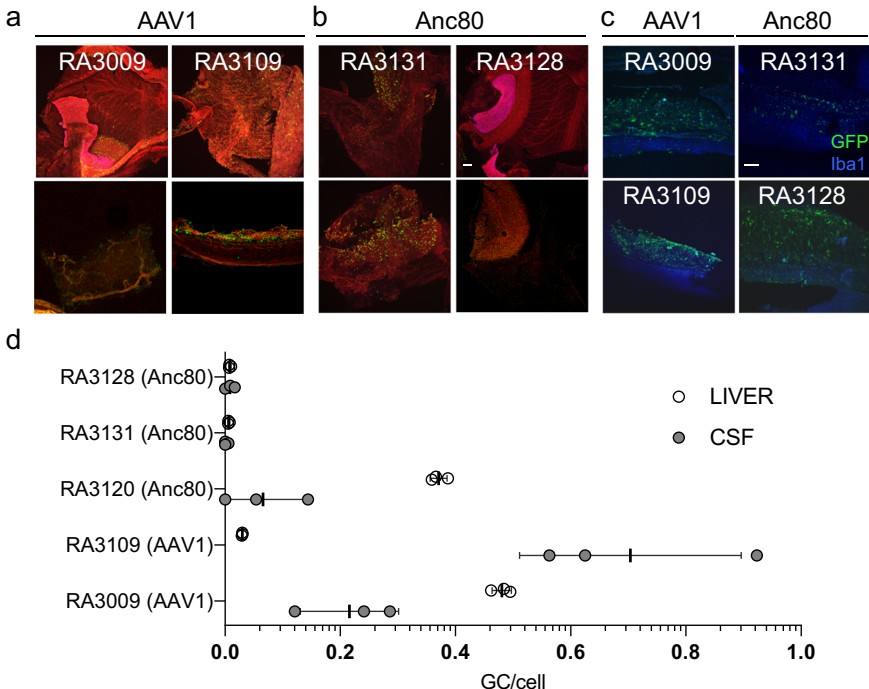

**Fig. 5 In vivo transduction of vestibular epithelia and other organs with AAV1 and Anc80L65 in rhesus macaques.** Representative whole mounts of the sensory epithelium from utricle, saccule, ampulla, and membranous semicircular ducts from animals that received **a** AAV1 ($n = 2$) or **b** Anc80L65 ($n = 2$). eGFP signal (green) was detected in three of the four animals that expressed eGFP; no expression was detected in animals RA3128 and RA3120. Red, phalloidin. Scale bar, 100 μm. **c** eGFP (green) expression was detected in the lateral wall along the length of the cochlea ($n = 2$ animals for Anc80L65 and $n = 2$ animals for AAV1); no differences were noticed between serotypes. Resident macrophages were stained with anti-Iba1 antibody. Scale bar, 100 μm. **d** Quantification of gene transfer of AAV1 and Anc80L65 carrying eGFP transgene in liver, spleen, and CSF following cochlear injection. Genomic DNA was harvested from liver, spleen, and CSF; eGFP genome copies (GC) per diploid cell were measured by droplet-digital PCR assay ($n = 3$ technical replicates of each tissue per animal). The data are shown as mean values ± standard deviation (SD). Source data are provided as a Source Data file.

based on biodistribution (copies detected in liver and spleen) and seroconversion (NAB), we concluded that the animal had received the vector. We can only hypothesize about lack of expression in this animal. Although the primate had some medical issues and was medicated prior to the injection, which may have impacted cell transduction[55], the observed result is most likely due to a delivery failure during the surgical procedure (unsuccessful stapes footplate fenestration) and/or failed intra-cochlear injection due to the small anatomical size and entailing limited exposure of the oval and round window. Yet, it is important to note that the injection procedure was well tolerated, with only one animal experiencing a mild unilateral facial paresis that just affected the eye closure and was related to the surgical transmastoid approach. In humans, the risk of facial nerve injury will be decreased given that the expected approach to access the round and oval window is through the external auditory canal route since the anatomical structures are larger. It is particularly exciting that most otologic surgeons will require little training to perform such intracochlear injections, as very similar approaches are used for other routine procedures, such as exploratory tympanotomy or cochlear implantation.

Taken together, our studies demonstrate efficient IHC targeting by AAV in a NHP model. These findings open the door to clinical evaluation of therapeutic gene transfer for forms of hearing loss affecting IHCs.

## Methods

**Viral vectors**. AAV2/1 and AAV2/Anc80L65 with a CMV- and CB7-driven eGFP transgene cassette were prepared at the Gene Transfer Vector Core (www.vdb-lab.org/vector-core) at Massachusetts Eye and Ear as previously described by HEK293 triple transfection[28]. All the vectors were triple titrated and tested for

purity and endotoxin levels (<0.125 EU/ml); AAV vectors were tested in mice prior to injection in NHP (Fig. 1 and Supplementary Fig. 1a).

Anc80L65 sequences are available at GenBank accession number AKU89595[28]. AAV2/Anc80L65 (ID 9230) plasmid reagents are available through http://www.addgene.com.

**Immunological assays**. Antibody titers against Anc80L65 in CSF and serum were determined through neutralization assays. Using a 96-well format, heat-inactivated CSF or serum samples (collected as described above) were serially diluted in serum-free medium (Life Technologies), and then treated with Anc80L65-luciferase ($10^7$ GC/well) for 1 h at 37 °C. The sample/Anc80L65-luciferase mix was then transferred onto HEK293 cells (10,000 cells/well), which had been treated with adenovirus (MOI 100) the day before. After 1 h at 37 °C, diluted serum medium (1 part serum-free, 2 parts serum) was added to each well. Two days later, the cells were treated with lysis buffer (Promega) and frozen at −80 °C for 30 min. The cells were then thawed at 37 °C for 15 min before being treated with substrate buffer (Tris-HCl, MgCl$_2$, ATP) (Life Technologies) and D-luciferin (Caliper Life Sciences). Luminescence output was read using the Synergy BioTek Plate Reader (BioTek). Data Analysis was performed using Microsoft Excel.

**Mouse experiments**. All experiments were approved by the Institutional Animal Care and Use Committee at Schepens Eye Research Institute (Protocol S-519-1021) as well as the Institutional Biosafety Committee (Protocols 13-068 and 18-121). The authors adhered to the NIH Guide for the Care and Use of Laboratory Animals. Animals were housed in ventilated Thoren rack cages. Mice were provided with food and water ad libitum. Mice were kept at temperature around 18–25 °C, 30–70% humidity and on 12-hour light/12-hour dark cycle. Wild-type C57BL/6J and pregnant CD1 mice were obtained from the Jackson Laboratory. For retro-orbital injections, C57BL/6 male mice (6-8 weeks) were anesthetized with Keta-mine/Xylazine intraperitoneally. Each animal was injected retro-orbitally (100 μl) with $1 \times 10^{11}$ GC/mouse of the following vectors: Anc80L65.CB7.eGFP.RBG and AAV1.CB7.eGFP.RBG. Animals were euthanized, and livers were collected and submerged in 4% paraformaldehyde (PFA) solution (Electron Microscopy Sciences) for 30 min, then placed in 30% sucrose overnight. The next day, each liver was mounted in Tissue-Tek O.C.T. Compound (Sakura Finetek) and flash frozen in cool isopentane. To visualize eGFP expression in liver, 15 μm sections were mounted with VECTASHIELD® Hard Set™ mounting medium with DAPI and

imaged with a Zeiss Axio Imager M2, at the same gain and intensity across all sections.

For mouse cochlear injections, 4-day-old male and female CD1 pups were injected via the round window membrane (RWM) using beveled glass microinjection pipettes. Pipettes were pulled from capillary glass (WPI) on a P-2000 pipette puller (Sutter Instrument) and were beveled (~20 μm tip diameter at a 28° angle) using a micropipette beveler (Sutter Instrument). Pups were anesthetized by rapid induction of hypothermia via immersion in ice/water for 2–3 min until loss of consciousness; this state was maintained on a cooling platform for 5–10 min during surgery. The surgical site was disinfected by scrubbing with betadine and wiping with 70% ethanol three times. A post-auricular incision was made to expose the transparent otic bulla. A micropipette was advanced manually through the bulla and overlying fascia and the RWM was penetrated by the tip of the micropipette. Approximately 0.75 μL of virus was injected unilaterally within 1 min into the left ear manually. Anc80L65.CB7.eGFP and Anc80L65.CMV.eGFP were used at a dose of $2 \times 10^9$ GC/μL. After the injection, the skin incision was closed using a 6-0 black monofilament suture (Surgical Specialties). Pups were subsequently returned to a 38 °C warming pad for 5–10 min and placed with their mothers for continued nursing. Mice were sacrificed 14 days later via transcardial perfusion with 4% PFA after induction of anesthesia with intraperitoneal administration of Ketamine/Xylazine. Inner ears were extracted, gently flushed with 4% PFA and left in this solution for an additional hour, then immersed in 0.12 M ethylenediaminetetraacetic acid (EDTA) for 48 h before being stored in phosphate buffered saline (PBS) until further processing. Subsequent microdissections were carried out to prepare cochlear and vestibular whole mounts that were stained with Green fluorescent protein (GFP [9F9.F9] #ab1218, Abcam, 1:500), Neurofilament H (#AB5539, Merck Millipore, 1:250) for neuronal structures and myosin 7A (Myo7A, #25-6790 Proteus Biosciences, 1:400) or phalloidin (#A22283, Thermo Fisher Scientific, 1:250) for hair cells in cochlear or vestibular tissue, respectively. After overnight incubation, the following secondary antibodies Goat anti-Mouse IgG1 Alexa Fluor 488 (#A-21121, Thermo Fisher Scientific 1:1000), Goat anti-Chicken IgY (H + L) Alexa Fluor 647 (#A-21449, Thermo Fisher Scientific 1:1000), and Goat anti-Rabbit IgG (H + L) Alexa Fluor 647 (#A-21245, Thermo Fisher Scientific 1:1000) were used prior to mounting of the specimens[22].

**Nonhuman primates.** Experiments with rhesus monkeys were performed at Schepens Eye Research Institute of Massachusetts Eye and Ear. All experimental procedures were approved by the Institutional Animal Care and Use Committees at Schepens Eye Research Institute (Protocol S484-0520) as well as the Institutional Biosafety Committee (Protocols 13–068 and 18–121). The authors adhered to the NIH Guide for the Care and Use of Laboratory Animals. Rhesus macaque monkeys were obtained from Covance; 3–4 year old female animals were used for experimentation. Animals were housed in pairs in stainless steel cages. Psychological/ environmental enrichment was provided. Target temperatures of 18–28 °C with a target relative humidity of 30–70%, and a 12-hour light/12-hour dark cycle was maintained. No statistical analysis between serotype transduction efficiencies was performed due to the limited access to specimens and qualitative nature of the reported findings.

**Phlebotomy and cerebrospinal fluid sampling.** Animals were sedated and the phlebotomy site was prepped with alcohol. Blood was obtained from the femoral or saphenous vein. A standard vacutainer needle was inserted into the vessel. Pressure was applied to the vessel after withdrawal of the needle to achieve hemostasis. NHP serum chemistry was performed by IDEXX BioAnalytics (North Grafton, MA, USA).

Cerebrospinal fluid (CSF) sampling from the cisterna magna was performed in a terminal procedure. Atropine was administered prior to the procedure. Once general anesthesia was established, the area was shaved and scrubbed. The immobilized animal's neck was held in flexion. A 1 ml CSF sample was collected using a 23G needle.

**Cochlear injection.** Atropine (0.05 mg/kg IM or SC) was administered prior to the surgical procedure. Once general anesthesia was established, the animal was given intravenous dexamethasone and cefazolin. The veterinary staff intubated the immobilized animal with a reinforced endotracheal tube and the tube was immobilized with a NeoBar ET tube holder (NeoTech). The left postauricular region was shaved and a semilunar incision was made sharply just posterior to the postauricular crease. A combination of soft and blunt dissection was used to follow a path just posterior to the external auditory canal until the hard bone of the mastoid cortex was encountered. Meticulous hemostasis was obtained using bipolar electrocautery. Once the mastoid cortex was encountered, a Lempert elevator was used to obtain broad exposure of the mastoid bone and self-retaining retractors were placed. The operating microscope was brought into place and used for the remainder of the procedure. High-speed otologic drills were used in combination with cutting and diamond burrs to perform mastoidectomy. The external auditory canal wall and the tympanic membrane were left intact. The facial recess was opened, and the facial nerve was skeletonized in the descending segment. Once the round window niche and stapes were visualized, a sharp surgical needle was used to

make a 2-mm fenestra in the oval window to allow fluid egress from the inner ear space during subsequent viral vector delivery through the round window membrane. Minimal to no perilymph leakage was observed prior the injection. The microinjection device (Harvard Apparatus) was primed, and a 33-gauge needle was gently inserted through the mastoid, facial recess, and the round window membrane to deliver the vector. A steady infusion of 30 μl of the viral vector was performed at a rate of 30 μL/min. The device was then withdrawn. The round window membrane and the fenestra in the stapes footplate were sealed with cohesive sodium hyaluronate viscoelastic (Healon® GV). All the animals were injected with $2.55 \times 10^{11}$ GC of viral vector in a total volume of 30 μl. Vector was diluted in sterile 0.9% NaCl solution and 0.001% Pluronic F68.

Supportive intravenous fluids were administered by the veterinary staff during the surgical procedures to compensate for any decrease in fluid consumption. The animals were monitored clinically for general well-being twice daily. They received buprenorphine for 72 hours after the surgery to prevent any pain and ondansetron as needed for nausea.

**Perfusion.** Atropine (0.05 mg/kg IM or SC) was administered prior to the procedure. Once general anesthesia had been established, animals were euthanized with Pentobarbital. Upon euthanasia, the anterior aspect of the neck was incised. The common carotid arteries and internal jugular veins were exposed bilaterally using blunt dissection. All four vessels were isolated and ligated at the proximal end using silk suture. One carotid artery and one jugular vein were also ligated distally to extend them. The other carotid artery and jugular vein were cut with iris scissors to accommodate size 4–5 French catheters. The catheters were inserted into the vessels approximately 3–4 cm and secured in place by tightening the distal suture. The catheter coming out of the jugular vein was placed into a waste container, and the carotid artery catheter was attached to a 60 ml syringe filled with formalin solution. The carotid artery was infused with 120 ml of formalin. Bilateral inner ears, CNS regions, and samples from different organs were harvested and processed for histology and/or biodistribution studies.

**Cochlear histological analysis.** Cochleae were fixed in 4% PFA and decalcified in 0.12M EDTA at room temperature with regular trimming of the bone for a month. Cochlear whole mounts were similarly prepared as previously reported[22]. However, sharp spring scissors (Fine Science Tools) were used to separate the lateral wall from the SC area (Supplementary Fig. 4b). The cochlear lateral wall was preserved and mounted separately (Supplementary Fig. 4d). Cochlear and vestibular whole mounts of ipsi- and contralateral ears were stained with antibodies against Green fluorescent protein (GFP [9F9.F9] #ab1218, Abcam, 1:500), myosin 7A (Myo7A, #25–6790 Proteus Biosciences, 1:200), phalloidin (#A22283, Thermo Fisher Scientific, 1:250), Iba1 (#019-19741, Wako Chemicals, 1:200) and Neurofilament H (#AB5539, Merck Millipore, 1:1000), together with corresponding secondary antibodies: Goat anti-Mouse IgG1 Alexa Fluor 488 (#A-21121, Thermo Fisher Scientific 1:1000), Goat anti-Chicken IgY (H + L) Alexa Fluor 568 (#A-11041, Thermo Fisher Scientific 1:1000), and Donkey anti-Rabbit IgG (H + L) Alexa Fluor 647 (#A-31573, Thermo Fisher Scientific 1:200). Mounting of the specimens was followed by epifluorescent and confocal microscopy.

The cochlear spiral frequency mapping microscopy was carried out using images acquired with an epifluorescence microscope equipped with a 10x objective. Photographs of all the pieces (at the same magnification) were obtained and imported into a composite image. Frequency maps in whole mounts were obtained after opening these images in ImageJ and subsequently using the plug-in "Measure_Line.class" downloaded from the Eaton-Peabody website, which semi-automatically calculates the frequencies at different points along the cochlea after manually tracing the organ of Corti of one ear from the most basal to the most apical piece. Once frequency locations were assigned, a confocal image of a given experimental series was obtained with the same settings, with laser intensity chosen based on the specimen with the strongest eGFP signal of all cochleae to prevent fluorescence saturation. For the quantification of eGFP-positive cells, an image obtained with the 63x objective of every mapped frequency region was manually quantified along the length of the cochlea by two independent researchers blinded to the treatment. The percentages were obtained by dividing the number of eGFP-positive cells by the total number of inner hair cells.

The areas of the spiral limbus and SCs were assessed qualitatively using a scale from "minus sign" (no expression) to "3 plus signs" (strongest signal). Control samples with secondary antibodies were used to exclude autofluorescence.

**Hematoxylin-eosin staining.** Samples from different tissues were fixed in 10% Normal Formalin Buffer. Tissues were paraffin embedded, sectioned, and stained with hematoxylin and eosin following standard protocols in the Histopathology Research Core at Massachusetts General Hospital.

**Tissue DNA biodistribution.** Snap frozen tissue was digested with proteinase K and genomic DNA (gDNA) was extracted using Blood & Cell Culture DNA Mini kit (Qiagen) as outlined in the manufacturer's protocol. Isolated gDNA was quantified, diluted and BamHI-digested, and genome copies (gc) distribution in diploid cells were quantified using the QX200 Droplet Digital PCR System and reagents (Bio-Rad) and transgene-specific primers/probes for eGFP (EGFP

Forward: AGCAAAGACCCCAACGAGAA, EGFP Reverse: GGCGGCGGTCAC-GAA, and EGFP Probe: 6FAM-CGCGATCACATGGTCCTGCTGG-TAMRA) multiplexed with the endogenous reference gene RNaseP assay (#4403326, Thermo Fisher Scientific). Data Analysis was performed using Microsoft Excel and GraphPad Prism 9.

**Reporting Summary**. Further information on research design is available in the Nature Research Reporting Summary linked to this article.

## Data availability

Source data analyzed are provided with this paper, cell (IHC and SC) quantifications and biodistribution data generated in this study are provided in the Source Data file. Raw confocal/microscopy images and datasets are available from the corresponding authors on reasonable request. Anc80L65 sequences are available at GenBank accession number AKU89595. AAV2/Anc80L65 (ID 9230) plasmid reagents are available through http://www.addgene.com. Source data are provided with this paper.

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

## Acknowledgements

We thank the Schepens Animal Care Facilities for support in the in vivo studies and the staff of the Gene Transfer Vector Core for vector production. Funding: Lonza Houston (L.H.V.), Giving/Grousbeck (L.H.V.); NIDCD grant R01DC015824 (K.M.S.), Nancy Sayles Day Foundation (K.M.S.), Lauer Tinnitus Research Center (K.M.S.), the Barnes Foundation (K.M.S.), the Zwanziger Foundation (K.M.S.), and Sheldon and Dorothea Buckler (K.M.S.).

## Author contributions

Conceptualization: E.A.M., L.D.L., C.U., W.F.S., M.J.M., K.M.S., L.H.V. Formal Analysis: E.A.M., L.D.L., C.U., K.M.S., L.H.V. Funding acquisition: K.M.S., L.H.V. Investigation: E.A.M., L.D.L., C.U., J.P., B.M.L., N.A.D., J.S., F.N., M.D.V., K.I.B., R.J.B., M.J.M., K.M.S.. Methodology: E.A.M., L.D.L., C.U., J.P., B.M.L., J.S., M.D.V., K.I.B., W.F.S., M.J.M., K.M.S., L.H.V. Project administration: E.A.M., CU. Supervision: E.A.M., M.J.M., K.M.S., L.H.V. Validation: E.A.M., C.U., L.H.V. Visualization: E.A.M., L.D.L., B.M.L., K.M.S. Writing—original draft: E.A.M., L.D.L., B.M.L., K.M.S. Writing—review & editing: E.A.M., L.D.L., C.U., J.S., M.J.M., K.M.S., L.H.V.

## Competing interests

L.H.V. is an inventor on several patents related to AAV gene therapy, including Anc80L65, AAV9, and method patents, which are licensed to several biopharma companies. During these studies, L.H.V. further receives funding from Lonza/Houston, Selecta Biosciences, and Solid Biosciences, which all had a licensing interest in the AncAAV technology. L.H.V. is a consultant and founder of Akouos, a hearing gene therapy company that has licensed the Anc80L65 technology. L.H.V. has a financial interest in Affinia, a company developing AAV gene therapies; he is an inventor of technology related to AAV gene therapy, a founder of the company, and also serves on its Board of Directors. L.H.V.'s interests were reviewed and are managed by M.E.E. and Partners HealthCare in accordance with their conflicts of interest policies. E.A.M., J.P., B.M.L., M.D.V. are employees at Akouos, Inc.; W.F.S. is a founder of Akouos, Inc.; M.J.M. is a founder and employee at Akouos, Inc. The other authors declare no competing interests.
