## [Peer Review File · Nature Communications]

Reviewers' Comments:

Reviewer #1:

Remarks to the Author:

This is a well written study describing the transduction of cochlear cells by AAV1 and Anc80 viral vectors in NHPs. The study findings show expression predominantly in inner hair cells, with only minor expression in outer hair cells and supporting cells, no expression in spiral ganglion neurons. It also documents viral spread to other organs and the presence of neutralising antibodies before and after injection. The findings will be of interest to those in the field. The methods were thorough and provided enough detail for the study to be replicated.

I found it surprising that there was little comparison to or discussion about other studies in NHP by Ivanchenko and Gyorgy. These papers were cited but there was little discussion about the differences in the approach or the results. Could you expand on these?

Other questions:

I believe that Healon is a liquid/gel-like substance. Given it was used to seal the fenestra in the oval window and round window after injection – do you believe that it would result in a lasting seal?

Was there any viral dose or perilymph leakage before, during or after viral injection?

What is the risk to the facial nerve with this approach? And what about cutting the chorda tympani? Given it's role in taste – did the animals eat normally afterwards?

Not much explanation is given for the lack of expression in RA3120? Is it not possible to visualise the injection very well?

Line 214: How was HC 'damage' assessed – do you mean minimal 'loss' of HCs?

There is not much comment in the discussion on the discrepancy between the lack of transduction in supporting cells and it's ability to transduce these cells in other species.

Line 349 – my understanding was that very few SCs were transduced, so why is it relevant to cite a gradient in morphological and metabolic differences in SCs?

Rachael Richardson

Reviewer #2:

Remarks to the Author:

Review

In this MS Andres-Mateos target cochlear gene transfer in Nonhuman Primates (NHP) in order to work towards clinical translation of inner ear gene therapy. They employed the clinically well established posterior tympanotomy for approaching the round window (as done for cochlear implant surgery) and combined administration of 30 µl of virus suspension with a vent at the oval window. Comparing AAV2/1 (2 animals) and AAV Anc80L65 (3 animals) they found Anc80L65, which they found highly potent for gene transfer into hair cells in mice in prior studies, to also work in NHPs. They studied transgenic expression of GFP as a readout of efficiency and specificity of the viral transduction in an early time window after administration (7-14 days following injection) and found that Anc80L65 in the best case transduced 90% of the inner hair cells. Routine veterinary observation and blood sampling revealed no adverse effects of the AAV administration. While neutralizing anti-AAV antibodies were found in the serum, they were not present in the cerebrospinal fluid.

This is work by two excellent groups and the work is timely and well executed.

This is one more important NHP study of inner ear gene therapy and encourages future clinical trials for inner hair cell targeted gene therapy with the Anc80 vector.

Limitations for informing future clinical trials arise from i) low sample size (2 and 3), ii) the observation time points being limited to a short window (7 days for 4 NHPs and 14 days for 1 NHP), iii) lack of assessing hearing before and after the procedure, iv) lack of a sham injection group, and v) the transgene being restricted to a small CDS (reporter).

Without measurements of auditory function, statements such as "AAV1 and Anc80L65 can be

delivered safely into the inner ear," seem hard to justify. Also, the biosafety part, which is important, is limited by the short observation time window. "Taken together, our studies demonstrate the safety and efficiency of IHC targeting in a NHP model." is an over-interpretation.

The MS preparation has shortcomings such as vague/qualitative reports such as:

"Overall, procedures went relatively uneventful and were well tolerated."

"Minimal hair cell damage was observed at the cochlear base where the injection was performed."

"The highest eGFP intensity was found at the apex, while the lowest intensity was observed at the base of the cochlea (Figure 3A)."

Moreover, some statements seem to lack the supporting evidence.

Also in cases, the referencing seems biased and has formatting issues (discussion).

For the discussion, I would deem it relevant to caution about the predictive power for expression of large CDS such as for OTOF where the expression in mouse IHCs clearly is lower than predicted based on transduction efficacy (e.g. Akil 2019; Al-Moyed 2019; Rankovic 2021). Moreover, while the quite specific inner hair cell expression (despite the broad promoter) is great and highly appreciated, commenting on supporting cell expression will be helpful for the reader. Finally, potential reasons for failing expression in animal RA3120 should be discussed.

Specific comments

Title:

Improved... vector: this vector has now been used for multiple preclinical studies on Cochlear Gene Transfer so I am not sure stressing the "improvement of vector" is fair.

Abstract:

In genetic mouse models of human disorders

Intro:

"between the ages of 65 and 74, approximately one in three people suffers it (3, 4)."

These references are on pediatric hearing impairment, please revise

"...for long-term expression (18)." In my reading, this study assessed safety but not long-term efficacy, from the paper: As previously reported, expression from the vector was subtherapeutic or limited in duration due to a cellular immune response to the AAV capsid, which prevented analysis of expression durability and long-term efficacy"

Please revise, potentially using reference to LCA2 gene therapy...

The referencing of preclinical work focuses heavily on hair cells and Harvard activities. It would be good to broaden including classical studies (e.g. Bedrosian et al., 2006), more recent studies of several capsids (e.g. Rankovic et al., 2021 where Anc80 seemed a bit inferior to PHP.eB in an overload AAV approach) as well as work targeting the spiral ganglion (e.g. Keppeler et al., 2018).

Results:

"AAV constructs were designed to include an optimized Chicken β -actin promoter with early CMV-enhancer (CB7) and chimeric intron (CI) given its prior use in NHP and human studies."

Please provide references

Data/Fig. 1: please provide titers in text or legend, the views on the tissue presented in B and D are not comparable, please provide better examples. Myo7a label is displaced in B/D

Data/Fig. 2:

"The fenestration allowed for injection of a larger volume without damage of the internal cochlear structure and enabled the vector to flow toward the apex of the cochlea."

Suggest to phrase this as reasoning rather than fact, if latter, do include data/references

"Frequency mapping of the injected cochleae was carried out using images acquired with an

epifluorescence microscope using a 10x objective.”
Please refer to methods and S4 on how tissue was prepared

Providing the total number of hair cells (e.g. right y-axis) will be helpful to provide the quantitative information on hair cell loss reported above.

Data/Fig. 4:

“We evaluated the eGFP expression in the lateral wall in all specimens obtained from the four animals where we discovered any eGFP-positive cells in the inner ear. eGFP signal was detected in all NHPs. Signal distribution and intensity were similar between all animals, and no differences were noticed between serotypes (Figure 5C). We did not detect colocalization between eGFP signal and microglia that was stained using an anti-Iba1 antibody and could not observe any differences in the Iba1 signal between Anc80L65 and AAV1”

This section is quite superficial, also what is the expectation for Iba1: colocalization would imply transduction of microglia?

Generally: when referring to representative images etc. please add number of observations.

Discussion:

See general comments. Avoid over-interpretation “safe procedure” or exaggerating “comprehensive examination of additional transduced cell types”, which would be expected to be backed by quantification (which is done for inner hair cells and supporting cells).

Please make sure to avoid readers to confuse eGFP intensity in cells (level) with transduction rate.

In my view the clinical ATOH1 trial used a similar surgical approach and should be cited.

Reviewer #3:

Remarks to the Author:

In this study, the authors examined the transduction of Anc80L65 and AAV1 in Rhesus inner ears, using the round window delivery route with oval window fenestration. They found that Anc80L65 was able to transduce the inner hair cells at higher rate compared to AAV1. They also showed that inner ear gene delivery with these two AAV serotypes caused minimal adverse side effects in NHPs, with the exception of one animal which had facial nerve paresis. My suggestions are listed below:

1. In the introduction, the authors stated that Anc80L65 is able to transduce 90% of IHCs in two animals. However, in Figure 3, the data shows that the IHC transduction rate is 90% only at the apex of the cochlea, and drops significantly as one moves toward the base. I find this statement a little misleading.
2. In the beginning of the results section, it is not clear to me what viral titers were used for the experiments in mice. The authors stated that the transduction rate for HCs are similar with CB7 promoter compared to CMV promoter. However, no quantification of the HC transduction was done. I would suggest the authors quantify the HC transduction rate between the AAVs with these two different promoters to support the statement they made.
3. In Figure 1B and 1D, it is not clear to me why these two figures are of different magnifications. It seems that they are not from the same specimen, according to the figure legend. If Fig 1B is with CMV promoter, and Fig 1D is with CB7 promoter, I would suggest that they are both shown with similar magnification so that the readers could compare the two more easily.
4. In table 1, it indicates that 4 out of 5 animals were euthanized 7 days after inner ear gene delivery, and only 1 animal was kept for 14 days after inner ear gene delivery. Yet, an important part of the study is on the presence of neutralizing antibodies. Is this enough of a time frame to assess the adaptive immune response to AAV? Typically adaptive immune response takes several weeks to mount. I wonder if the neutralization assay data may be an underestimate given the short time course between gene delivery and euthanasia.
5. In Figure 3b, the authors did not plot the data for RA3120, since there was no cochlear HC

transduction in this animal. However, I would suggest that the authors include this data in the figure. It is my opinion that this would give the readers a more complete picture of the data.

6. It seems to me that there may be some IHC and OHC loss in some of the specimens in Figure 3A. Perhaps a comparison with the contralateral non-injected ear would be helpful to assess for HC loss? The authors also stated in the discussion that the advantage of the oval window fenestration is the prevention of HC loss. I am not sure this is the case from Figure 3A. A useful way to assess for inner ear damage would be to assess the auditory function on these animals before and after inner ear gene delivery.

7. Figure 4 is difficult for me to understand. Where in the cochlea are these images taken from? They seem to have very different magnifications. What do the different colors represent?

8. I also find Figure 5 to be difficult to understand. What are the differences between the top and bottom rows of Figures 5a and 5b? Which vestibular organs were these images taken from? Was quantification of the vestibular HC transduction done for comparison? Figure 5c shows the lateral wall, but these images look very different from one another. Which portion of the cochlea were these specimens taken from? What kind(s) of cells in the lateral wall were transduced with the AAVs?

9. The figure legend of Figure 5 seems to have some mistakes. The authors wrote "representative whole mounts of the sensory epithelium from the ampulla, utricle, saccule and semicircular canal from animals...", this seems to suggest that the ampulla is a separate and different structure independent of the semicircular canals. Also, the authors wrote that "eGFP signal was detected in three of the four animals" that received Anc80L65, yet according to the methods section, only three animals received Anc80L65.

10. In the discussion, the authors wrote that "only one animal experiencing a mild facial paresis". I am not sure I would consider this a mild complication. Did this animal recover the facial function or was the animal euthanized prior to full recovery?

11. Table S2, the data for RA3128 has a line separating the Day 0 and Day 7 columns, which is different from all other animals' data. Is this intentional?

In Figure S2 and in the results section, it would be interesting to include in the text what type(s) of supporting cells were transduced by each of the AAVs

Reviewer #1 (Remarks to the Author):

This is a well written study describing the transduction of cochlear cells by AAV1 and Anc80 viral vectors in NHPs. The study findings show expression predominantly in inner hair cells, with only minor expression in outer hair cells and supporting cells, no expression in spiral ganglion neurons. It also documents viral spread to other organs and the presence of neutralizing antibodies before and after injection. The findings will be of interest to those in the field. The methods were thorough and provided enough detail for the study to be replicated.

We thank the reviewer for the supportive remarks.

I found it surprising that there was little comparison to or discussion about other studies in NHP by Ivanchenko and Gyorgy. These papers were cited but there was little discussion about the differences in the approach or the results. Could you expand on these?

We agree that these papers deserve more discussion. We added the following paragraph after the in-text citations of the mentioned papers:

“Namely, Gyorgy et al.¹¹ included data of one male cynomolgus monkey with 92% of IHCs and OHCs eGFP transduction seven weeks after delivery of 3×10^{11} vector genomes (VG) of AAV9-PHP.B.CBA.eGFP in 10 μ L. In contrast, a female macaque injected bilaterally with the same vector and same volume at dose 1×10^{11} VG showed limited cochlear transduction eight weeks after dosing. In both animals, the vector was delivered by injection through the round window membrane. An additional three animals were injected with the same vector mentioned above by bilateral intracochlear injection with three different doses⁴⁰. According to the authors, nearly all of IHCs and OHCs were transduced from base to apex at the two higher doses (3.5 and 7×10^{11} VG), while there was a steep reduction in vector transduction at the lower dose (2×10^{11} VG). Nevertheless, only cochlear cross-sections were examined and quantified in these studies, which do not allow a direct and equal comparison with our cochlear whole mount data quantification. Expanding on these studies, the same group recently injected three cynomolgus monkeys with a new capsid variant AAV-S⁴¹. Although the two highest doses (4.7 and 5.8×10^{11} VG) showed extensive levels of eGFP expression in a variety of cell types in the NHP cochlea and vestibular system, unfortunately, the analysis in NHP was limited to the AAV-S variant. Hence, comparisons cannot be made with other AAV capsids to more extensively characterize them in the cochlea.”

Other questions:

I believe that Healon is a liquid/gel-like substance. Given it was used to seal the fenestra in the oval window and round window after injection – do you believe that it would result in a lasting seal?

We have included additional information in the methods. In our studies, Healon GV was used, which is a cohesive-type viscoelastic composed of sodium hyaluronate with very high viscosity. It may create a temporary seal until the RWM completely heals/seals. It is routinely used by some physicians during cochlear implant surgery (covering the round window/coating of the electrode prior to insertion) and has originally been described by Lehnhardt [Intracochlear placement of cochlear implant electrodes in soft surgery technique. HNO. 1993 <https://pubmed.ncbi.nlm.nih.gov/8376183/>]. As outlined by Friedland and Runge-Samuelson, who provide detailed references to substantiate their conclusions (Soft Cochlear Implantation: Rationale for the Surgical Approach. Trends Amplif. 2009 <https://www.ncbi.nlm.nih.gov/pmc/articles/PMC4111526/#bibr59-1084713809336422>), the

“use of a hyaluronate-based lubricant in fact appears to be beneficial to promoting hearing preservation when opening the inner ear. This may be secondary to cytostatic properties of the hyaluronate, reduced friction and trauma during electrode placement, prevention of perilymph leakage, and/or prevention of cochlea contamination with blood and bone dust.” To summarize, although we are not aware of any post-mortem study looking at the integrity of the round window after opening it with or without Healon, everything points towards a temporary seal that is being created to prevent perilymph leakage and cochlear contamination.

Was there any viral dose or perilymph leakage before, during or after viral injection?

As stated now in the methods, minimal to no perilymph leakage was observed prior the injection. During the vector administration (injection with 30 μ L), due to the expected fluid displacement, some leakage was observed around the oval window. We did not observe what is known as a “gusher” prior, during, or after injection in any of the animals and are therefore confident that, beyond the fluid displacement expected during the injection, the vast majority of the solution remained in the cochlea for a substantial amount of time.

What is the risk to the facial nerve with this approach? And what about cutting the chorda tympani? Given its role in taste – did the animals eat normally afterwards?

The surgical access via posterior tympanotomy is the same as for any routine cochlear implantation. Thus, the risk to the facial nerve and the chorda tympani would be comparable to what is expected during this common procedure. Assuming that the reviewer’s questions are focused on translational efforts, temporary dysgeusia could occur in up to a third of patients, but permanent chorda tympani dysfunction is only expected in 1-2% of individuals (Jeppesen et al., <https://www.ncbi.nlm.nih.gov/pmc/articles/PMC3793264/>). Nevertheless, food and water intake was monitored twice daily during the study, and we did not observe any changes in the animals’ food intake after the surgery.

With respect to the facial nerve, the surgery was more challenging in NHP compared to humans due to the small size of the temporal bones and cochleae (comparable to a human newborn or smaller). Nonetheless all the anatomical landmarks necessary for a successful surgical transmastoid approach were recognizable, and only one of the animals experienced a mild facial paresis that resulted in incomplete eye closure ipsilateral to the intracochlear injection. It did not hamper food and water intake, and no behavioral changes were noticed. Hence, from a translational standpoint, we expect that the surgery will be less complex in humans (as explained in the discussion). A transcanal approach to the round window will likely be the preferred approach for human patients and the risk to compromise those two nerves would be even lower.

Not much explanation is given for the lack of expression in RA3120? Is it not possible to visualise the injection very well?

A more extensive discussion has been included in the manuscript: “We can only hypothesize about lack of expression in this animal. Although the primate had some medical issues and was medicated prior to the injection, which may have impacted cell transduction {Chai, 2019 #52}, the observed result is most likely due to a delivery failure during the surgical procedure (unsuccessful stapes footplate fenestration) and/or failed intracochlear injection due to the small anatomical size and entailing limited exposure of the oval and round window.”

Line 214: How was HC ‘damage’ assessed – do you mean minimal ‘loss’ of HCs?

The HC damage was assessed by analyzing the lack of the Myo7A immunofluorescent signal in certain areas where HCs would have been expected. While this in general is seen as more precise than a Phalloidin stain to determine the integrity of hair cells, it is still not ideal and we cannot automatically conclude that all of these hair cells are ‘lost’.

There is not much comment in the discussion on the discrepancy between the lack of transduction in supporting cells and its ability to transduce these cells in other species. Line 349 – my understanding was that very few SCs were transduced, so why is it relevant to cite a gradient in morphological and metabolic differences in SCs?

The last two comments are answered together as there must have been a misunderstanding. We hope the revised introduction is clearer. While hardly any eGFP-positive outer hair cells were detected, the supporting cell transduction was substantial in these monkeys, i.e., comparable to other species. Fig. S2 shows that the highest qualitative level of transduction (+++) was reached in 21 out of 68 possible positions (assessed in total: 17 frequencies x 4 animals). The representative image for this category depicts that these are not very few supporting cells. Figure 3 can be misleading in this regard as it is focusing on the hair cells. Because the eGFP expression of the inner hair cells is so intense, one might miss the supporting cell signal (e.g., see RA3131, Middle).

Reviewer #2 (Remarks to the Author):

Review

In this MS Andres-Mateos target cochlear gene transfer in Nonhuman Primates (NHP) in order to work towards clinical translation of inner ear gene therapy. They employed the clinically well established posterior tympanotomy for approaching the round window (as done for cochlear implant surgery) and combined administration of 30 μ l of virus suspension with a vent at the oval window. Comparing AAV2/1 (2 animals) and AAV Anc80L65 (3 animals) they found Anc80L65, which they found highly potent for gene transfer into hair cells in mice in prior studies, to also work in NHPs. They studied transgenic expression of GFP as a readout of efficiency and specificity of the viral transduction in an early time window after administration (7-14 days following injection) and found that Anc80L65 in the best case transduced 90% of the inner hair cells. Routine veterinary observation and blood sampling revealed no adverse effects of the AAV administration. While neutralizing anti-AAV antibodies were found in the serum, they were not present in the cerebrospinal fluid.

This is work by two excellent groups and the work is timely and well executed.

This is one more important NHP study of inner ear gene therapy and encourages future clinical trials for inner hair cell targeted gene therapy with the Anc80 vector.

Limitations for informing future clinical trials arise from i) low sample size (2 and 3), ii) the observation time points being limited to a short window (7 days for 4 NHPs and 14 days for 1 NHP), iii) lack of assessing hearing before and after the procedure, iv) lack of a sham injection group, and v) the transgene being restricted to a small CDS (reporter).

We agree with the reviewer that additional studies need to be done to inform future clinical trials. However, these additional studies are beyond the scope of the current paper. In the meantime, we believe our data will promote further preclinical development to bring inner ear gene therapy to the clinic.

Without measurements of auditory function, statements such as “AAV1 and Anc80L65 can be delivered safely into the inner ear,” seem hard to justify. Also, the biosafety part, which is important, is limited by the short observation time window. “Taken together, our studies demonstrate the safety and efficiency of IHC targeting in a NHP model.” is an over-interpretation.

The MS preparation has shortcomings such as vague/qualitative reports such as:
“Overall, procedures went relatively uneventful and were well tolerated.”
“Minimal hair cell damage was observed at the cochlear base where the injection was performed.”
“The highest eGFP intensity was found at the apex, while the lowest intensity was observed at the base of the cochlea (Figure 3A).”
Moreover, some statements seem to lack the supporting evidence.
Also in cases, the referencing seems biased and has formatting issues (discussion).
We thank the reviewer for all the insightful comments. We have amended the manuscript to avoid data overinterpretation, and to clarify previously vague and inaccurate statements throughout the manuscript.

For the discussion, I would deem it relevant to caution about the predictive power for expression of large CDS such as for OTOF where the expression in mouse IHCs clearly is lower than predicted based on transduction efficacy (e.g. Akil 2019; Al-Moyed 2019; Rankovic 2021). Moreover, while the quite specific inner hair cell expression (despite the broad promoter) is great and highly appreciated, commenting on supporting cell expression will be helpful for the reader. Finally, potential reasons for failing expression in animal RA3120 should be discussed.
Additional text has been added to the results and discussion sections to address these comments.

Specific comments

Title:

Improved... vector: this vector has now been used for multiple preclinical studies on Cochlear Gene Transfer so I am not sure stressing the “improvement of vector” is fair.
We have modified the title to address the legitimate concern
“Choice of Vector and Surgical Approach Enables Efficient Cochlear Gene Transfer in Nonhuman Primate”

Abstract:

In genetic mouse models of human disorders
We thank the reviewer for noticing this mistake, which we have corrected in the revised manuscript.

Intro:

“between the ages of 65 and 74, approximately one in three people suffers it (3, 4).”
These references are on pediatric hearing impairment, please revise
We corrected the references.

“...for long-term expression (18).” In my reading, this study assessed safety but not long-term efficacy, from the paper: As previously reported, expression from the vector was subtherapeutic or limited in duration due to a cellular immune response to the AAV capsid, which prevented analysis of expression durability and long-term efficacy”
Please revise, potentially using reference to LCA2 gene therapy...
We have replaced the reference.

The referencing of preclinical work focuses heavily on hair cells and Harvard activities. It would be good to broaden including classical studies (e.g. Bedrosian et al., 2006), more recent studies of several capsids (e.g. Rankovic et al., 2021 where Anc80 seemed a bit inferior to

PHP.eB in an overload AAV approach) as well as work targeting the spiral ganglion (e.g. Keppeler et al., 2018).

We appreciate the suggestion, and we have included the references in the revised manuscript.

Results:

“AAV constructs were designed to include an optimized Chicken β -actin promoter with early CMV-enhancer (CB7) and chimeric intron (CI) given its prior use in NHP and human studies.”

Please provide references

We have provided the references.

Data/Fig. 1: please provide titers in text or legend, the views on the tissue presented in B and D are not comparable, please provide better examples. Myo7a label is displaced in B/D

We thank the reviewer for this comment. We have included the doses injected to the mice and corrected the labels in the figure. The experimental details are also mentioned in the methods section.

Because a direct comparison of promoters in mice was never the focus of this study, we only qualitatively concluded that similar tropism and transduction efficiency was observed between the two promoters. We agree with the reviewer that the presentation of two magnifications for different specimens might be confusing at first, but because the Landegger et al. Nature Biotechnology 2017 paper had already included several overview images of the vestibular system, we wanted to present some new details to the readers (even though these obviously were different animals than the ones included in the 2017 study). On the right below, please see an overview image of a semicircular canal crista of one of the mice injected with the CMV promoter. On the left, the image of the mouse injected with the CB7 promoter (already included in the manuscript). Again, the goal of these experiments was not to directly compare the promoters and determine that one is better than the other, but rather to verify that CB7 works in the (murine) ear in general, so we could proceed with the monkey studies.

Data/Fig. 2:

“The fenestration allowed for injection of a larger volume without damage of the internal cochlear structure and enabled the vector to flow toward the apex of the cochlea.”

Suggest to phrase this as reasoning rather than fact, if latter, do include data/references

We have re-phrased our statement.

“Frequency mapping of the injected cochleae was carried out using images acquired with an

epifluorescence microscope using a 10x objective.”

Please refer to methods and S4 on how tissue was prepared

A more detailed explanation has been included in the methods.

Providing the total number of hair cells (e.g. right y-axis) will be helpful to provide the quantitative information on hair cell loss reported above.

Outer hair cell loss/damage was not quantified, but inner hair cells were quantified, and no loss or damage was detected. Since no loss was observed with either AAV1 or Anc80L65 and the figure is focused on the efficient transduction, we have not included the numbers in the figure. But we included the following paragraph: “Outer hair cell damage was not quantified, but sporadic outer hair cell damage was observed in injected and contralateral uninjected ears for all the animals. Inner hair cells were quantified at each frequency (a total of 20 cells per frequency region along the length of the cochlea) and no cell damage was observed with either serotype (data non included). Minimal tissue damage at the most basal portion of the cochlea was detected in a few specimens, most likely related to the specimen post-mortem dissection or the injection procedure performed.”

Data/Fig. 4:

“We evaluated the eGFP expression in the lateral wall in all specimens obtained from the four animals where we discovered any eGFP-positive cells in the inner ear. eGFP signal was detected in all NHPs. Signal distribution and intensity were similar between all animals, and no differences were noticed between serotypes (Figure 5C). We did not detect colocalization between eGFP signal and microglia that was stained using an anti-Iba1 antibody and could not observe any differences in the Iba1 signal between Anc80L65 and AAV1”

This section is quite superficial, also what is the expectation for Iba1: colocalization would imply transduction of microglia?

We changed this section.

Generally: when referring to representative images etc. please add number of observations.

We added the number of animals.

Discussion:

See general comments. Avoid over-interpretation “safe procedure” or exaggerating “comprehensive examination of additional transduced cell types”, which would be expected to be backed by quantification (which is done for inner hair cells and supporting cells).

We have removed all the comments as suggested by the reviewer.

Please make sure to avoid readers to confuse eGFP intensity in cells (level) with transduction rate.

We have made the appropriate changes in the manuscript.

In my view the clinical ATOH1 trial used a similar surgical approach and should be cited. The ATOH1 clinical trial also used an intra-labyrinthine (IL) delivery route, but the injection was performed through the stapes footplate without fenestration. Although both routes of delivery are similar, the injection through the stapes footplate (oval window) targets the scala vestibuli and the injection through the round window targets the scala tympani, hence vector tropism and distribution in the inner ear will not be directly comparable.

Reviewer #3 (Remarks to the Author):

In this study, the authors examined the transduction of Anc80L65 and AAV1 in Rhesus inner ears, using the round window delivery route with oval window fenestration. They found that Anc80L65 was able to transduce the inner hair cells at higher rate compared to AAV1. They also showed that inner ear gene delivery with these two AAV serotypes caused minimal adverse side effects in NHPs, with the exception of one animal which had facial nerve paresis. My suggestions are listed below:

1. In the introduction, the authors stated that Anc80L65 is able to transduce 90% of IHCs in two animals. However, in Figure 3, the data shows that the IHC transduction rate is 90% only at the apex of the cochlea, and drops significantly as one moves toward the base. I find this statement a little misleading.

We corrected the statement as suggested by the reviewer.

2. In the beginning of the results section, it is not clear to me what viral titers were used for the experiments in mice. The authors stated that the transduction rate for HCs are similar with CB7 promoter compared to CMV promoter. However, no quantification of the HC transduction was done. I would suggest the authors quantify the HC transduction rate between the AAVs with these two different promoters to support the statement they made.

We thank the reviewer for this comment. We have included the dose in figure 1. Also, the experimental details are mentioned in the methods section.

We agree that the quantification of HC transduction rate in the injected mice would be relevant. However, because a direct comparison of promoters in mice was never the focus of this study, we only qualitatively concluded that there were no major differences between the two promoters. Yet, we prepared a table that summarizes our findings and hopefully will lead the reviewer to the same conclusion:

Inner hair cells:

Mouse	Most basal	-				Most apical
CB7-1	-	-	-	-	-	-
CB7-2	++	+++	+++	+++	+++	+++
CB7-3	+++	+++	+++	+++	+++	+++
CMV-1	+++	N/A (dried)	+++	+++	+++	+++
CMV-2	+++	+++	+++	+++	+++	+++
CMV-3	+++	+++	+++	+++	+++	+++

Outer hair cells:

Mouse	Most basal	-				Most apical
CB7-1	-	-	-	-	-	-
CB7-2	+	+++	+++	+++	+++	+++
CB7-3	++	+++	+++	+++	+++	+++
CMV-1	+++	N/A (dried)	+++	+++	+++	+++
CMV-2	+++	+++	+++	+++	+++	+++
CMV-3	+++	+++	+++	+++	+++	+++

Table S3. Qualitative assessment of *in vivo* transduction of inner and outer hair cells of CD1 mice 14 days after P4 ipsilateral round window membrane delivery of Anc80L65 using the same eGFP transgene driven by CMV or CB7 (n=3 mice each). The hair cell transduction from most basal (left) to most apical (right) was rated using a scale from “+++” (strongest signal) to “-“ (no expression). Identical confocal microscope settings were used to obtain all

images. No obvious differences were detected between the promoters, but the injection of one animal (CB7-1) likely was not completely successful. In addition, one specimen of CMV-1 could not be assessed due to difficulties during the processing (dried while mounting, thus no analysis of fluorescence possible).

3. In Figure 1B and 1D, it is not clear to me why these two figures are of different magnifications. It seems that they are not from the same specimen, according to the figure legend. If Fig 1B is with CMV promoter, and Fig 1D is with CB7 promoter, I would suggest that they are both shown with similar magnification so that the readers could compare the two more easily.

We agree with the reviewer that the presentation of two magnifications for different specimens might be confusing at first, but because the Landegger et al. Nature Biotechnology 2017 paper had already included several overview images of the vestibular system, we wanted to present some new details to the readers (even though these obviously were different animals than the ones included in the 2017 study). On the right below, please see an overview image of a semicircular canal crista of one of the mice injected with the CMV promoter. On the left, the image of the mouse injected with the CB7 promoter (already included in the manuscript). Again, the goal of these experiments was not to directly compare the promoters and determine that one is better than the other, but rather to verify that CB7 works in the (murine) ear in general, so we could proceed with the monkey studies.

4. In table 1, it indicates that 4 out of 5 animals were euthanized 7 days after inner ear gene delivery, and only 1 animal was kept for 14 days after inner ear gene delivery. Yet, an important part of the study is on the presence of neutralizing antibodies. Is this enough of a time frame to assess the adaptive immune response to AAV? Typically adaptive immune response takes several weeks to mount. I wonder if the neutralization assay data may be an underestimate given the short time course between gene delivery and euthanasia.

We agree with the reviewer that neutralizing antibodies may be underestimated. Based on AAV preclinical and clinical experience NAb titers start to go up during the first week after dosing, peaking between 2 and 6 weeks. Despite the NAb titers potentially being higher at later time-points, our overall conclusion that all the animals seroconverted will not change.

5. In Figure 3b, the authors did not plot the data for RA3120, since there was no cochlear HC transduction in this animal. However, I would suggest that the authors include this data in the figure. It is my opinion that this would give the readers a more complete picture of the data.

We have included the data in the figure and figure legend.

6. It seems to me that there may be some IHC and OHC loss in some of the specimens in Figure 3A. Perhaps a comparison with the contralateral non-injected ear would be helpful to assess for HC loss? The authors also stated in the discussion that the advantage of the oval window fenestration is the prevention of HC loss. I am not sure this is the case from Figure 3A. A useful way to assess for inner ear damage would be to assess the auditory function on these animals before and after inner ear gene delivery.

We thank the reviewer for this important comment. We have observed such sporadic hair cell loss (or rather lack of Myosin7A staining) in both injected and contralateral ears to the same extent. Possible explanations include dissection artifacts or age-related damage (as the macaques were young adults). Due to the bilateral damage, we would not overinterpret the sporadic lack of Myo7A staining in Figure 3A, but agree that we should rephrase the advantages of oval window fenestration and have thus removed the statement regarding prevention of HC loss.

We agree with the reviewer that ABRs and DPOAEs will be very useful in a future study, beyond the scope of the current work. The purpose of this study was to assess the short-term feasibility of the procedure and AAV transduction, so auditory function was not measured because it would have required repeated episodes of general anesthesia in a short period of time. In addition, early post-surgery auditory function will be markedly affected by the post-surgical trauma.

7. Figure 4 is difficult for me to understand. Where in the cochlea are these images taken from? They seem to have very different magnifications. What do the different colors represent?

Figure 4 is focused on the spiral ganglion. During the preparation of the cochlear whole mounts, cuts through the modiolus (parallel to the cochlear turns) were made and these sections were mounted and stained. The images are all obtained with a 20x objective, but the size of the modiolar cuts and thus the spiral ganglion neuron area is variable. Hence, the images appear to have different magnifications, but when comparing the cell bodies of the spiral ganglion neurons, one can see that this is not the case. In summary, what the figure is trying to demonstrate is the lack of spiral ganglion neuron transduction. If we had expression in the neurons, we would see a dual expression of eGFP (green) and neurofilament (red/pink) as the latter stains neuronal structures.

8. I also find Figure 5 to be difficult to understand. What are the differences between the top and bottom rows of Figures 5a and 5b? Which vestibular organs were these images taken from? Was quantification of the vestibular HC transduction done for comparison? Figure 5c shows the lateral wall, but these images look very different from one another. Which portion of the cochlea were these specimens taken from? What kind(s) of cells in the lateral wall were transduced with the AAVs?

We agree with the reviewer that Figure 5 is not as straightforward to understand as for example Figure 3. Because this is among the first papers that analyze the cellular AAV transduction in non-human primates with whole mounts and not just cross sections, we had to establish the respective tissue processing techniques to obtain representative samples. This worked very well for the cochlear parts of the inner ear, but was more challenging for the vestibular organs. Thus, Figures 5a and 5b show different representative images of several areas of vestibular tissue including utricle (top image for RA3009 and RA3128), semicircular canals (RA3109 bottom), and the vestibule. No quantification of vestibular hair cell transduction was carried out.

Whole mounts of the lateral wall (as dissected in Figure S4, panel D) were evaluated for eGFP expression. Although we observed variability between animals, fluorescence was detected in at least 3 to 4 sections of the 7 pieces dissected (along the length of the cochlea), mainly in the spiral ligament/stria vascularis. We did not characterize cells and areas in the lateral wall that were transduced with AAV since the sensory epithelium was prioritized for whole mounts, and cochlear cross-sections will be more appropriate for this type of analysis and the characterization of the anatomical location. While it is possible to easily repeat experiments with mice or other rodents, this is exponentially more difficult with non-human primates given the limited number of monkeys that can be used for an experiment.

9. The figure legend of Figure 5 seems to have some mistakes. The authors wrote “representative whole mounts of the sensory epithelium from the ampulla, utricle, saccule and semicircular canal from animals...”, this seems to suggest that the ampulla is a separate and different structure independent of the semicircular canals. Also, the authors wrote that “eGFP signal was detected in three of the four animals” that received Anc80L65, yet according to the methods section, only three animals received Anc80L65.

We thank the reviewer for this comment and have clarified the figure legend as follows: “Representative whole mounts of the sensory epithelium from utricle, saccule, ampulla, and membranous semicircular ducts from...” However, we believe that the other statement is correct as we wrote “eGFP signal (green) was detected in three of the four animals that expressed eGFP; no expression was detected in animal RA3128.” Specifically, five animals were injected (3x Anc80, 2x AAV1) and four animals expressed eGFP (all except RA3128). We have modified the text to “no expression was detected in animals RA3128 and RA3120”.

10. In the discussion, the authors wrote that “only one animal experiencing a mild facial paresis”. I am not sure I would consider this a mild complication. Did this animal recover the facial function or was the animal euthanized prior to full recovery?

We agree with the reviewer that it is debatable what is considered a minor/mild or major surgical complication. We defined the paresis as “mild” since it was unilateral, only affected the eyelid closure, and did not affect the lower face. Additional language has been included in the manuscript for clarification. As we opted for a partial tarsorrhaphy, we couldn’t evaluate if this animal fully recovered the facial nerve function prior to euthanasia; but it also wouldn’t be expected with the relatively short observation period as these injuries usually take at least several weeks to subside.

11. Table S2, the data for RA3128 has a line separating the Day 0 and Day 7 columns, which is different from all other animals’ data. Is this intentional?

We thank you the reviewer for noting the inconsistency and we have modified the table.

In Figure S2 and in the results section, it would be interesting to include in the text what type(s) of supporting cells were transduced by each of the AAVs

We thank the reviewer for this comment, agree, and are also very interested in supporting cells. However, due to the limited number of channels for the immunofluorescence and the many crucial stains that we had to incorporate, we found it very difficult to pursue additional immunofluorescent staining with antibodies like Sox2 or Cytokeratin 19 that would have helped us to differentiate the various types of supporting cells. However, what can be said is that the transduction showed no gradient from immediately next to the third outer hair cell row to the most lateral cells preserved prior to the transition to the lateral wall. Thus, as can be seen in Figure 3 (quantification in Fig. S2), Hensen’s and Claudius’ cells seem to be transduced to the same extent. However, what also can be seen in Figure 3 is that there is no eGFP expression in the area of Deiters’ cells, phalangeal or pillar cells.

ADDITIONAL SUGGESTED CHANGES PRIOR TO RESUBMISSION:

Fig. 1:

Myo7A between B and D is not in the same line as the other labels any more.

We have corrected the figure.

Fig. S2 legend:

The areas of the spiral limbus and supporting cells were assessed qualitatively using a scale from “3” (strongest signal) to “-“ (no expression).

In the actual Figure S2, we have “+++”, not “3”, so I would change the legend to:

The areas of the spiral limbus and supporting cells were assessed qualitatively using a scale from “+++” (strongest signal) to “-“ (no expression).

We appreciate the careful review of the manuscript and we have addressed the inconsistencies in the figure.

Reviewers' Comments:

Reviewer #1:

Remarks to the Author:

I believe the authors have thoroughly addressed the questions and issues of the reviewers and I have no further comments to raise.

Reviewer #2:

Remarks to the Author:

The authors have revised the MS by editing the text. They have better balanced the statements, references and toned down the claims. There are no new data/analysis included.

I frankly don't understand why the authors would like to refrain from providing a cytochrome c oxidase (COX) immunohistochemistry for the inner hair cells or at least densities/# cells/100 μ m for select frequency regions. "Inner hair cells were quantified at each frequency (a total of 20 cells per frequency region along the length of the cochlea) and no cell damage was observed with either serotype (data non included)."

Overall, the MS has been improved.

Reviewer #3:

Remarks to the Author:

The authors have adequately addressed my concerns. It is my opinion that this revised manuscript should be accepted for publication.

REVIEWERS' COMMENTS

Reviewer #1 (Remarks to the Author):

I believe the authors have thoroughly addressed the questions and issues of the reviewers and I have no further comments to raise.

We thank you the reviewer for all the comments and suggested edits that contributed to improve the quality of the manuscript.

Reviewer #2 (Remarks to the Author):

The authors have revised the MS by editing the text. They have better balanced the statements, references and toned down the claims. There are no new data/analysis included.

I frankly don't understand why the authors would like to refrain from providing a cytochleogram for the inner hair cells or at least densities/# cells/100 μ m for select frequency regions. "Inner hair cells were quantified at each frequency (a total of 20 cells per frequency region along the length of the cochlea) and no cell damage was observed with either serotype (data non included)."

Overall, the MS has been improved.

We appreciate the time and effort that the reviewer has dedicated to providing valuable feedback on the manuscript. Although an assessment of HC cell survival was done along the cochlea for all the animals in the study, no formal blinded HC quantification / cytochleograms was performed during the study. Given that the paper was focus on the optimization, the feasibility and tolerability of AAV gene therapy in the cochlea, the authors did not include the quantifications. In addition, no differences were found between serotypes.

Reviewer #3 (Remarks to the Author):

The authors have adequately addressed my concerns. It is my opinion that this revised manuscript should be accepted for publication.

We are grateful to the reviewer for their insightful comments on the paper.